# Adhesion toughness of multilayer graphene films

Joseph D. Wood [1,2], Christopher M. Harvey [1] & Simon Wang [1,3]

Interface adhesion toughness between multilayer graphene films and substrates is a major concern for their integration into functional devices. Results from the circular blister test, however, display seemingly anomalous behaviour as adhesion toughness depends on number of graphene layers. Here we show that interlayer shearing and sliding near the blister crack tip, caused by the transition from membrane stretching to combined bending, stretching and through-thickness shearing, decreases fracture mode mixity $G_{II}/G_I$, leading to lower adhesion toughness. For silicon oxide substrate and pressure loading, mode mixity decreases from 232% for monolayer films to 130% for multilayer films, causing the adhesion toughness $G_c$ to decrease from 0.424 J m$^{-2}$ to 0.365 J m$^{-2}$. The mode I and II adhesion toughnesses are found to be $G_{Ic} = 0.230$ J m$^{-2}$ and $G_{IIc} = 0.666$ J m$^{-2}$, respectively. With point loading, mode mixity decreases from 741% for monolayer films to 262% for multilayer films, while the adhesion toughness $G_c$ decreases from 0.543 J m$^{-2}$ to 0.438 J m$^{-2}$.

[1] Department of Aeronautical and Automotive Engineering, Loughborough University, Loughborough, Leicestershire LE11 3TU, UK. [2] Department of Mechanical Engineering, Imperial College London, London SW7 2AZ, UK. [3] School of Mechanical and Equipment Engineering, Hebei University of Engineering, Handan 056038, China. Correspondence and requests for materials should be addressed to J.D.W. (email: joseph.wood@imperial.ac.uk) or to C.M.H. (email: c.m.harvey@lboro.ac.uk) or to S.W. (email: s.wang@lboro.ac.uk)

Koenig et al.[1] suggested that one possible cause for the large decrease in adhesion toughness for multilayer graphene films in comparison to monolayer ones is the roughness of the substrate surface. Multilayer graphene films may conform less well to the substrate than monolayer ones. Koenig et al.[1] made roughness measurements on the top surfaces of graphene films and found a large drop in roughness from monolayer to two layer; however, they also found a large drop from two layer to three layer. This suggests that the roughness of the substrate surface is unable to explain the large decrease in adhesion toughness. To investigate the effect of interface roughness further, Gao and Huang[2] argued that the rough surface of silicon oxide causes graphene films to bend; hence, the total adhesion energy consists of both van der Waals interaction energy and a negative contribution of bending strain energy. By assuming the substrate to have a sinusoidal rough surface, they attempted to calculate the adhesion energy. They concluded that the large decrease in adhesion toughness from monolayer to multilayer graphene films is due to the increase in bending strain energy caused by the large increase in the bending stiffness. Jiang and Zhu[3] measured the van der Waals interaction energy between monolayer graphene films and silicon oxide substrate using atomic force microscopy. Their measurements show, however, that the roughness increases the interaction energy. In contrast, He et al.[4] studied the large decrease in adhesion toughness from another perspective. They proposed that the total adhesion energy consists of both van der Waals interaction energy and residual in-plane strain energy due to lattice mismatch strain at the graphene film-silicon oxide interface. Their results show that the van der Waals interaction energy remains nearly the same for graphene films with any number of layers, but that the residual in-plane strain energy and Young's modulus decrease sharply from monolayer to multilayer graphene films. Koenig et al.[1], however, reported convincing experimental results that show a constant Young's modulus. This observation provided a solid foundation for their subsequent adhesion toughness calculations using a continuum mechanics approach.

Koenig et al.[1] also suggested possible sliding between graphene layers in multilayer graphene films. The present work follows Koenig et al.'s[1] continuum mechanics approach but with consideration for the interlayer shearing and sliding effect. Furthermore, the present work considers the effect of shearing and sliding on the fracture mode mixity. This is an important consideration, since interface adhesion toughness is not a purely intrinsic material property, but instead also depends on the mode mixity.

Note that the fracture mode mixity and the interlayer shear and sliding effect are not considered anywhere in the current analytical mechanical models[1–10] and we argue that this has caused confusion when calculating adhesion toughness. Cao et al.[11,12] did, however, recently report studies on adhesion toughness between photoresist films and copper substrates using blister tests and the finite element method. Two types of film are considered:

One is pure photoresist film and the other is combined photoresist film and a monolayer graphene. Mode mixity is considered by using cohesive zone modelling.

The present work shows that adhesion toughness is mode mixity dependent, and that interlayer shearing and sliding near the blister crack tip, caused by the transition from membrane stretching to combined bending, stretching and through-thickness shearing, decreases the mode mixity $G_{II}/G_I$, consequently reducing the adhesion toughness $G_c$. By considering the interlayer shearing and sliding effect, the mode I and mode II toughnesses are shown to be independent of the number of graphene layers. Accounting for the interlayer shearing and sliding effect on the fracture mode mixity explains the behaviour reported in the literature[1], where adhesion toughness measurements seemingly depend on the film thickness (i.e., the number of graphene layers). Once the mode I and mode II adhesion toughnesses have been found, the linear failure criterion can accurately determine the adhesion toughness under general loading conditions for real-world applications of graphene film-substrate systems.

## Results

**Circular blister test under a pressure load**. Figure 1 shows two types of circular blister test to determine the adhesion toughness of mono- and multilayer graphene films. The blister has a crack tip radius $R_B$, the thickness of the monolayer graphene is $t$, $n$ represents the number of graphene layers and the Young's modulus of graphene is $E$. In Fig. 1a, the blister is under pressure loading[1]. According to Jensen[13,14], the deflection $\delta$ at the centre of the blister in the membrane limit is

$$\delta = f(\nu)\left(\frac{pR_B^4}{nEt}\right)^{1/3} \tag{1}$$

in which $p$ is the pressure load and $f(\nu)$ is given by Storåkers[15] as

$$f(\nu) = 0.9635\left[\frac{3(1-\nu)}{7-\nu}\right]^{1/3} \tag{2}$$

The coefficient of 0.9635 in Eq. (2) is introduced in the present work to achieve the benchmark value of $f(1/3) = 0.645$ obtained by Jensen[13] since Storåkers' formula[15] $f(\nu) = [3(1-\nu)/(7-\nu)]^{1/3}$ is approximate. The bending moment per unit width $M_B$, in-plane force per unit width $N_B$, and shear force per unit width $P_B$, at the blister crack tip[13,14] can be expressed in the following forms,

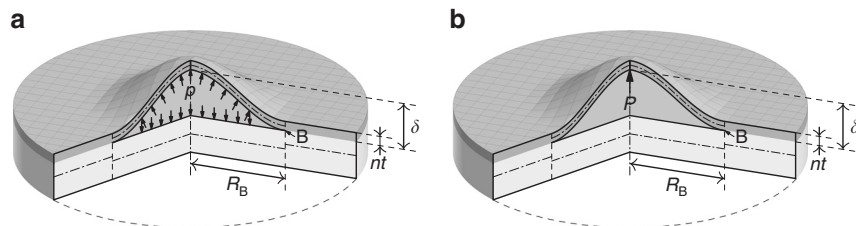

**Fig. 1** Circular blister tests to determine the adhesion toughness of mono- and multilayer graphene films. **a** A blister under a pressure load $p$. **b** A blister under a point load $P$

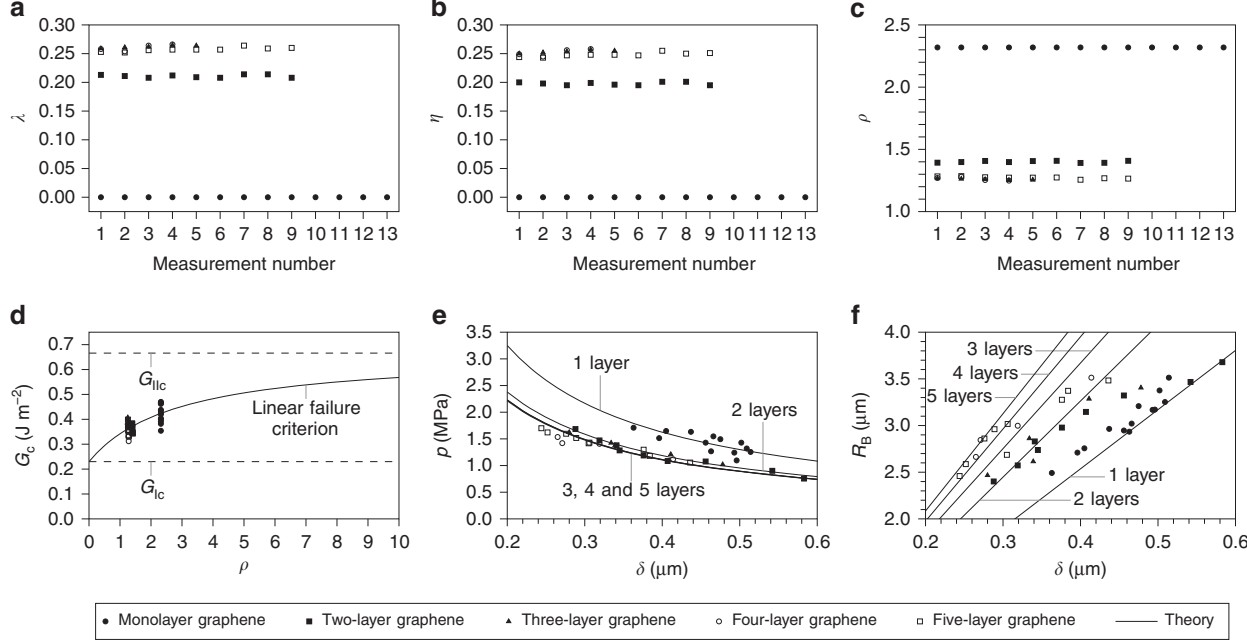

**Fig. 2** Delaminating graphene films under a pressure load. **a–c** Plots showing calculated values of the interlayer shearing and sliding parameter $\lambda$ (**a**), the ratio $\eta = G_S/G_J$ (**b**) and the fracture mode mixity $\rho = G_{II}/G_I$ (**c**) based on the measured values of $p$ and $\delta$, and the material properties of monolayer graphene. **d** Plot showing adhesion toughness $G_c$ vs. the fracture mode mixity $\rho$. **e, f** Plots showing the measured and theoretical relationships between the pressure load $p$ (**e**) and the blister radius $R_B$ (**f**) vs. the deflection at the centre of the blister $\delta$

respectively (Supplementary Fig. 2):

$$M_B = \frac{nt}{4}\left[\frac{nEtp\delta}{3(1-\nu^2)\varphi(\nu)f(\nu)}\right]^{1/2} \quad (3)$$

$$N_B = \left[\frac{nEtp\delta}{f(\nu)}\right]^{1/2}\varphi(\nu) \quad (4)$$

$$P_B = \frac{1}{2}pR_B \quad (5)$$

in which the Poisson's ratio $\nu$-dependent parameter $\varphi(\nu)$ is

$$\varphi(\nu) = \frac{(1.078 + 0.636\nu)^{2/3}}{2[6(1-\nu^2)]^{1/3}} \quad (6)$$

At this stage, the effect of interlayer shearing and sliding on the fracture mode mixity can be introduced. An introduction to mixed-mode partition theory is given in Supplementary Note 1. This theory is then developed and extended for the thin film blister test in Supplementary Note 2. The mode I and II energy release rates (ERRs) are obtained as[16–20]

$$G_I = 0.6227 \times \frac{p\delta}{8}\frac{(0.7578 - 0.1429\nu + \lambda)^2}{\varphi(\nu)f(\nu)} \quad (7)$$

$$G_{II} = 0.3773 \times \frac{p\delta}{8}\frac{(1.400 + 0.2358\nu)^2}{\varphi(\nu)f(\nu)} \quad (8)$$

and the mode mixity ratio $\rho = G_{II}/G_I$ as

$$\rho = 0.6059\left(\frac{1.400 + 0.2358\nu}{0.7578 - 0.1429\nu + \lambda}\right)^2 \quad (9)$$

The $\lambda$ parameter in Eqs. (7) and (9) represents the interlayer

shearing and sliding effect at the blister crack tip, which is given as

$$\lambda = \overline{\lambda}S(n) \quad (10)$$

By using Eq. (1) and Supplementary Eq. (50) in conjunction with mixed-mode partition theory[16–20], the parameter $\overline{\lambda}$ in Eq. (10) can have the following alternative expressions:

$$\overline{\lambda} = \overline{\zeta}(\nu)\left(\frac{pR_B}{nEt}\right)^{1/3} = \overline{\zeta}(\nu)\frac{1}{f(\nu)}\frac{\delta}{R_B} = \overline{\zeta}(\nu)\left(\frac{p\delta}{f(\nu)nEt}\right)^{1/4} \quad (11)$$

where

$$\overline{\zeta}(\nu) = 3.442\left[(1-\nu^2)\varphi\right]^{1/2} \quad (12)$$

In the case of monolayer graphene films, the shear force in Eq. (5) makes no contribution to the ERR in the membrane limit because there is no interlayer shearing and sliding. In the case of multilayer graphene films, interlayer shearing and sliding occurs near the blister crack tip, caused by the transition from membrane stretching to combined bending, stretching and through-thickness shearing. Consequently, interlayer shearing and sliding activates the shear force in Eq. (5). Its action is introduced through the $\lambda$ parameter in conjunction with the interlayer shearing and sliding factor $S(n)$, which is assumed to take the following form:

$$S(n) = 1 - e^{1-n} \quad (13)$$

A more thorough and detailed explanation for the origin of $\lambda$ is given in Supplementary Note 2.

The total ERR is simply the sum of the mode I ERR $G_I$ in Eq. (7) and the mode II ERR $G_{II}$ in Eq. (8). The mode mixity-dependent adhesion toughness $G_c$ can now be determined by using the mode I and mode II adhesion toughnesses and a linear failure criterion in which $G_c = (1 + \rho)/(1/G_{Ic} + \rho/G_{IIc})$. Note that $G_{Ic}$ and $G_{IIc}$ are intrinsic interface material properties but $G_c$ is not. One major aim of the present study is to determine

**Table 1 Average adhesion toughness of multilayer graphene films**

| | $G_J$ (J m$^{-2}$) | | $G_c$ (J m$^{-2}$) | | $\rho = G_{II}/G_I$ | |
|---|---|---|---|---|---|---|
| | Present mechanical model | Koenig et al.[1] | Present mechanical model | Koenig et al.[1] | Present mechanical model | Koenig et al.[1] |
| Monolayer | 0.424 | 0.450 | 0.424 | 0.450 | 2.319 | 2.320 |
| Multilayer | 0.295 | 0.310 | 0.365 | 0.310 | 1.299 | 2.320 |

values for $G_{Ic}$ and $G_{IIc}$ based on Koenig et al.'s[1] experimental results. Once these two properties are known, the adhesion toughness under other loading conditions can be readily calculated.

The total ERR, which includes the contributions from the crack tip bending moment $M_B$ in Eq. (3), the in-plane force $N_B$ in Eq. (4), and the crack tip shear force $P_B$ in Eq. (5), can also be written in terms of the $G_J$ component from Jensen's work[13,14], which does not account for the interlayer shearing and sliding effect, and the additional interlayer shearing and sliding component from the present work $G_s$, as follows:

$$G = G_J + G_S = G_J(1 + \eta) \tag{14}$$

Jensen's $G_J$ component can be calculated as[13,14]

$$G_J = \zeta(\nu)\left(\frac{p^4 R_B^4}{nEt}\right)^{1/3} = \zeta(\nu)\frac{nEt}{f^4(\nu)}\left(\frac{\delta}{R_B}\right)^4 = \zeta(\nu)\frac{p\delta}{f(\nu)} \tag{15}$$

in which the parameter $\zeta$ is

$$\zeta(\nu) = \frac{1}{8\varphi} + \frac{(1-\nu^2)\varphi^2}{2} \tag{16}$$

The ratio $\eta = G_S/G_J$ is

$$\eta = \frac{\lambda(\lambda + 1.516 - 0.2858\nu)}{1.761 + 0.1835\nu + 0.05413\nu^2} \tag{17}$$

Koenig et al.[1] found that $Et = 347$ N m$^{-1}$ with $E \approx 1$ TPa. Taking Poisson's ratio $\nu = 0.16$ (following ref. [1]), then Eqs. (2), (6), (12) and (16) give $f(0.16) = 0.6907$, $\varphi(0.16) = 0.3099$, $\overline{\zeta}(0.16) = 1.891$ and $\zeta(0.16) = 0.4502$, respectively. Then, the essential equations above, namely Eqs. (15), (17), (11) and (9), become, respectively

$$G_J = 0.4502\left(\frac{p^4 R_B^4}{nEt}\right)^{1/3} = 1.978nEt\left(\frac{\delta}{R_B}\right)^4 = 0.6517p\delta \tag{18}$$

$$\eta = 0.5580\lambda(1.470 + \lambda) \tag{19}$$

$$\overline{\lambda} = 1.891\left(\frac{pR_B}{nEt}\right)^{1/3} = 2.738\frac{\delta}{R_B} = 2.075\left(\frac{p\delta}{nEt}\right)^{1/4} \tag{20}$$

$$\rho = \frac{1.252}{(0.7349 + \lambda)^2} \tag{21}$$

Note that Koenig et al.[1] used $G_J = 0.655\,p\delta$, which is very close to Eq. (18) in the present work. Furthermore, by combining either Eqs. (8) and (15), or Eqs. (19), (21) and $G_{II}(1 + 1/\rho) = G_J(1 + \eta)$, then

$$G_{II} = 0.6986G_J \tag{22}$$

In the following, the pressure $p$, the central deflection $\delta$ and the radius $R_B$ of the multilayer graphene film blisters are taken from figures in Koenig et al.'s[1] Supplementary Information. The results are presented in Fig. 2. In Fig. 2a–c, the calculated values of $\lambda$, $\eta$

and $\rho$, respectively, for monolayer and multilayer graphene films are plotted based on the measured values of $p$ and $\delta$ from Koenig et al.[1]. In Fig. 2d, the calculated adhesion toughness $G_c$ is plotted vs. the fracture mode mixity $\rho$. In Fig. 2e, f, comparisons are made between the measured values of $p$, $\delta$ and $R_B$, and the present mechanical model for graphene films with different numbers of layers. Note that the 'Theory' curve in Fig. 2e is obtained by substituting Eqs. (7) and (8) into the linear failure criterion and solving for $p$; then for Fig. 2f, use of Eq. (1) recasts the theory in terms of $R_B$ and $\delta$. There is generally very good agreement between the present mechanical model and the experimental measurements[1].

The numerical data for Fig. 2 is also recorded in Supplementary Tables 1–5 for mono-, two-, three-, four- and five-layer graphene film blisters, respectively. To keep consistency with Koenig et al.[1], results are calculated using the pressure $p$ and the central deflection $\delta$ meaning that $G_J = 0.6517\,p\delta$ and $\overline{\lambda} = 2.075[p\delta/(nEt)]^{1/4}$ from Eqs. (18) and (20) are the forms that used. For the purpose of completeness and comparison, results have also been calculated using the alternative expressions for $\overline{\lambda}$ in Eq. (20), namely $\overline{\lambda} = 1.891[pR_B/(nEt)]^{1/3}$ and $\overline{\lambda} = 2.738\delta/R_B$. The results are presented in Supplementary Tables 6–10 and Supplementary Tables 11–15, respectively. There is generally good agreement between the results when using the different expressions for $\overline{\lambda}$. The values of the $\lambda$ parameter, based on Koenig et al.'s[1] measurements, are recorded in Supplementary Tables 1–15. There is a large decrease from monolayer to two-layer graphene films and then only a small decrease from two-layer to three-layer graphene films. For the three-, four- and five-layer graphene films, the values of the $\lambda$ parameter are very close to each other. This shows the typical interlayer shearing and sliding behaviour.

The average adhesion toughnesses are $G_c = 0.424$, 0.362, 0.389, 0.348 and 0.359 J m$^{-2}$ for the mono-, two-, three-, four- and five-layer graphene film blisters, respectively, which correspond to the following mode mixities $\rho = G_{II}/G_I = 2.319$, 1.400, 1.259, 1.263 and 1.272. There is a large decrease in mode mixity for two-layer graphene film blisters in comparison to monolayer films, which results in a large decrease in the adhesion toughness. For higher numbers of graphene layers, the adhesion toughness does not change significantly from the two-layer case as there are no significant changes in mode mixity. An overall average adhesion toughness for multilayer graphene films blisters is $G_c = 0.365$ J m$^{-2}$ with $\rho = G_{II}/G_I = 1.299$. These results are shown in Table 1.

Now the mode I and mode II adhesion toughnesses, $G_{Ic}$ and $G_{IIc}$, are considered. He et al.[4] showed that the van der Waals interaction energy remains nearly the same for graphene films with any number of layers at 0.266 J m$^{-2}$. This suggests that $G_{Ic}$ and $G_{IIc}$ are the same for interfaces between monolayer graphene films and silicon oxide substrates, and between multilayer graphene films and silicon oxide substrates. As adhesion toughness is generally very small, a linear failure criterion can provide an accurate representation of the fracture mechanics in question[18].

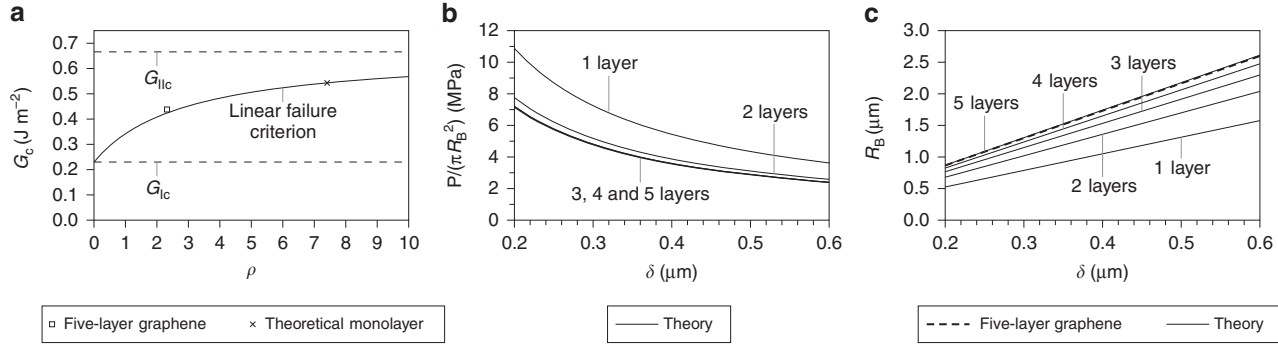

**Fig. 3** Delaminating graphene films under a point load. **a** Plot showing adhesion toughness $G_c$ vs. the fracture mode mixity $\rho$. **b, c** Plots showing the theoretical relationships between the point load $P$ (**b**) and the blister radius $R_B$ (**c**) vs. the deflection at the centre of the blister $\delta$. Note that the average measured value of $\delta/R_B = 0.2309$ for five-layer graphene is also shown

Let subscripts '1' and '2+' represent values for monolayer and multilayer graphene films, respectively. Substituting the monolayer and multilayer results from Table 1 into the linear failure criterion and solving simultaneously gives

$$G_{Ic} = \frac{G_{c1}G_{c2+}(\rho_1 - \rho_{2+})}{\rho_1 G_{c1}(1 + \rho_{2+}) - \rho_{2+}G_{c2+}(1 + \rho_1)} = 0.230 \, \text{J m}^{-2} \quad (23)$$

and

$$G_{IIc} = \frac{G_{c1}G_{c2+}(\rho_1 - \rho_{2+})}{G_{c2+}(1 + \rho_1) - G_{c1}(1 + \rho_{2+})} = 0.666 \, \text{J m}^{-2} \quad (24)$$

It is interesting to note that $G_{Ic} = 0.230 \, \text{J m}^{-2}$ is very close to He et al.'s[4] theoretical calculation of the van der Waals interaction energy at $0.266 \, \text{J m}^{-2}$. In fact, the van der Waals interaction energy is essentially the same in concept as the mode I adhesion toughness. The mode I adhesion toughness $G_{Ic}$ can be determined using atomic force microscopy measurements[3] and JKR model as

$$G_{Ic} = \frac{2F_{adh}}{3\pi R_{tip}} = 0.198 \, \text{J m}^{-2} \quad (25)$$

where $F_{adh} = 378 \, \text{nN}$ is the van der Waals interaction force and $R_{tip} = 405.4 \, \text{nm}$ is the radius of the microsphere tip used in the atomic force microscopy measurements. It is seen that the measured $G_{Ic} = 0.198 \, \text{J m}^{-2}$ is very close to the present value of $G_{Ic} = 0.230 \, \text{J m}^{-2}$.

In the following section, the theory developed above for the circular blister test under pressure loading and the determined values of $G_{Ic} = 0.230 \, \text{J m}^{-2}$ and $G_{IIc} = 0.666 \, \text{J m}^{-2}$ will be used to predict adhesion toughness under point loading in order to examine the validity of the approach.

**Circular blister test under a point load.** A blister under a point load (refs. [10,13]) is shown in Fig. 1b. The mechanical model for it is very similar to the model developed above for a pressure load. Some essential formulae are recorded here. Fitting a curve to the data in Jensen's[13] Fig. 15 gives $\varphi(\nu)$ as

$$\varphi(\nu) = 0.382\nu^3 + 0.013\nu^2 + 0.248\nu + 0.422 \quad (26)$$

The function $f(\nu)$ now becomes

$$f(\nu) = 1/(2\varphi(\nu)) + 2\varphi^2(\nu)(1 - \nu^2) \quad (27)$$

The pressure load $p$ can now simply be replaced everywhere with $P/(\pi R_B^2)$. By making this substitution in Eqs. (3) to (5), the

mode I and II ERRs can be obtained as[16–20]

$$G_I = 0.6227 \times \frac{P\delta}{8\pi R_B^2} \frac{\left(1 - 1.557\sqrt{(1 - \nu^2)\varphi^3} + \lambda\right)^2}{\varphi(\nu)f(\nu)} \quad (28)$$

$$G_{II} = 0.3773 \times \frac{P\delta}{8\pi R_B^2} \frac{\left(1 + 2.569\sqrt{(1 - \nu^2)\varphi^3}\right)^2}{\varphi(\nu)f(\nu)} \quad (29)$$

In addition, Eqs. (11) and (15) become, respectively

$$\overline{\lambda} = \overline{\zeta}(\nu)\left(\frac{P}{\pi R_B nEt}\right)^{1/3} = \overline{\zeta}(\nu)\frac{1}{f(\nu)}\frac{\delta}{R_B} = \overline{\zeta}(\nu)\left(\frac{P\delta}{\pi R_B^2 f(\nu)nEt}\right)^{1/4} \quad (30)$$

$$G_J = \zeta(\nu)\left(\frac{P^4}{\pi^4 R_B^4 nEt}\right)^{1/3} = \zeta(\nu)\frac{nEt}{f^4(\nu)}\left(\frac{\delta}{R_B}\right)^4 = \zeta(\nu)\frac{P\delta}{\pi R_B^2 f(\nu)} \quad (31)$$

Taking Poisson's ratio $\nu = 0.16$, then Eqs. (26), (27), (12) and (16) give $\varphi(0.16) = 0.4636$, $f(0.16) = 1.497$, $\overline{\zeta}(0.16) = 2.313$ and $\zeta(0.16) = 0.3743$, respectively. Equations (28)–(31) then produce the following:

$$G_J = 0.3743\left(\frac{P^4}{\pi^4 R_B^4 nEt}\right)^{1/3} = 0.07446nEt\left(\frac{\delta}{R_B}\right)^4 = 0.25\frac{P\delta}{\pi R_B^2} \quad (32)$$

$$\eta = 0.4485\lambda(1.030 + \lambda) \quad (33)$$

$$\overline{\lambda} = 2.313\left(\frac{P}{\pi R_B nEt}\right)^{1/3} = 1.545\frac{\delta}{R_B} = 2.091\left(\frac{P\delta}{\pi R_B^2 nEt}\right)^{1/4} \quad (34)$$

$$\rho = \frac{1.9640}{(0.5149 + \lambda)^2} \quad (35)$$

$$G_{II} = 0.8809G_J \quad (36)$$

From Eq. (35), it can be seen that $\rho = 7.407$ for monolayer graphene under a point load, which is much larger than for the pressure loading condition at $\rho = 2.319$. The adhesion toughness for monolayer graphene under a point load can be estimated using $G_{Ic} = 0.230 \, \text{J m}^{-2}$, $G_{IIc} = 0.666 \, \text{J m}^{-2}$ and a linear failure criterion to be $G_c = 0.543 \, \text{J m}^{-2}$, which is clearly larger than for the pressure loading case at $G_c = 0.424 \, \text{J m}^{-2}$.

The adhesion toughness for multilayer graphene under point loading can be estimated in a similar way as above for pressure loading but now using experimental data from Zong et al.[5] in which they used nanoparticles to create a point load on five-layer graphene membrane blisters. The blisters typically possessed a radius $R_B$ in the range 250–300 nm and central deflection $\delta$ in the range 50–70 nm. They used the formula $G_c = 0.0625nEt(\delta/R_B)^4$ with $E = 0.5$ TPa and $nt = 1.7$ nm. Note that Zong et al.'s[5] value for $E$ is half of that used by Koenig et al.[1], and that $n \approx 5$. Zong et al. reported the adhesion toughness as $G_c = 0.151$ J m$^{-2}$ meaning that $\delta/R_B = 0.2309$. When using Koenig et al.'s[1] value of $E = 1.0$ TPa, then Eq. (32) gives $G_J = 0.360$ J m$^{-2}$, and Eq. (14) gives the total measured adhesion toughness as $G_c = 0.438$ J m$^{-2}$. Now using $\rho = 2.624$ from Eq. (35), the linear failure criterion, and the mode I and mode II adhesion toughnesses, $G_{Ic} = 0.230$ J m$^{-2}$ and $G_{IIc} = 0.666$ J m$^{-2}$, the predicted value of $G_c$ is $G_c = 0.437$ J m$^{-2}$, which is extremely close to measured $G_c = 0.438$ J m$^{-2}$.

It can be seen that the mode mixity plays a key role in determining the adhesion toughness and that the accuracy of $G_{Ic} = 0.230$ J m$^{-2}$, $G_{IIc} = 0.666$ J m$^{-2}$ and the linear failure criterion is very good.

Figure 3 shows the behaviour of delaminating graphene films under a point load. Figure 3a–c follows the same style as Fig. 2d–f; however, the measured data[5] is now only for films with five layers. In particular, it is seen in Fig. 3c that the measured value of $\delta/R_B = 0.2309$ is very close to the theoretical prediction of $\delta/R_B = 0.2298$.

## Discussion

In recent work[21,22] (following ref. [1]), Boddeti et al. reported further studies on the adhesion toughness between monolayer graphene and silicon oxide substrates. The adhesion toughness was found to be $G_c = 0.24$ J m$^{-2}$, which is significantly smaller than $G_c = 0.45$ J m$^{-2}$, reported by Koenig et al.[1]. Boddeti et al.[21] suggest that the difference arises from the differences in interface properties such as roughness and chemical reactivity between the samples in ref. [1] and the samples in refs. [21,22]. In line with this suggestion, the present work suggests that the reduction is caused by reduction of the mode I and mode II adhesion toughnesses at the interface, $G_{Ic}$ and $G_{IIc}$, which are now estimated. Taking the Young's modulus and Poisson's ratio still as $E = 1$ TPa and $\nu = 0.16$, Eq. (22) gives the mode II ERR components at failure as $G_{II} = 0.6986 \times 0.24 = 0.168$ J m$^{-2}$. Then the mode I ERR component at failure is easily obtained as $G_I = 0.072$ J m$^{-2}$. If the ratio between $G_{Ic}$ and $G_{IIc}$ is taken to be the same as that in ref. [1], i.e., $G_{IIc}/G_{Ic} = 2.896$, then $G_{Ic}$ and $G_{IIc}$ are then calculated to be $G_{Ic} = 0.130$ J m$^{-2}$ and $G_{IIc} = 0.377$ J m$^{-2}$. Clearly they are significantly smaller than $G_{Ic} = 0.230$ J m$^{-2}$ and $G_{IIc} = 0.666$ J m$^{-2}$ for the samples in ref. [1]. More information on adhesion toughness of graphene can be found in the latest review paper[23].

The methodology developed above is also applied in the authors' recent work (manuscript in review) to determine the mode I and mode II adhesion toughness of thin films by using blister tests. The analytical predictions agree very well with the experimental results reported by Cao et al.[11].

It should be noted that a general methodology has been presented, and the substrate should not be restricted to silicon oxide substrates. Furthermore, the 'adhesion energy' commonly used in the literature is generally different from the adhesion toughness unless the mode I adhesion toughness is equal to mode II adhesion toughness, which is not generally the case. It is the adhesion toughness that matters for the design of graphene film-substrate material systems.

**Data availability**. The authors declare that the data supporting the findings of this study are available within the article and its Supplementary Information file.

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

## Acknowledgements

This work was supported by the UK Engineering and Physical Sciences Research Council (EPSRC) under grant reference EP/M000958/1.

## Author contributions

S.W. and C.M.H. conceived the ideas. All authors contributed to the theory derivations. J.D.W. performed the calculations and prepared the manuscript draft. S.W. and C.M.H. checked all of J.D.W.'s calculations, and prepared the final draft of the manuscript. S.W.

and C.M.H. jointly supervised the whole project. All authors participated in discussions about the project.

## Additional information

**Competing interests:** The authors declare no competing financial interests.

