## [Peer Review File · Nature Communications]

Reviewers' Comments:

Reviewer #1:

Remarks to the Author:

In this manuscript, Wood et al. analytically studied the adhesion between graphene (monolayer and multilayer) and substrate with continuum mechanics approach based on a blister model. By considering fracture model mixity and sliding effect in multilayer graphene membranes, the authors explained the drop of interfacial adhesion from monolayer to multilayer graphene, as observed in previous experiment (Nature Nanotechnol, 6, 543–6, 2011). By fitting this experimental data to the analytical model, the authors calculated the intrinsic mode I and mode II adhesion toughness for graphene/SiO₂ interface and then applied these values to study the blister under a point load.

The novelties claimed in this work are model mixity and sliding effects for graphene/substrate adhesion analysis. While the model mixity, as a basic ideal in interfacial fracture problems, has been considered for the adhesion of graphene (for example, Int. J. Solids and Structures 84, 147-159, 2016), up to now, the sliding effects in multilayer graphene membrane has not been considered for interfacial adhesion.

Sliding is a reasonable consideration since the van der Waals interaction between graphene layers may induce interlayer relative displacement under delamination between graphene and substrate. However, the authors need to correlate the physical mechanism of sliding to the parameter λ in the manuscript, which was introduced as a dimensionless parameter indicating the level of sliding. Moreover, the authors need to confirm the values of λ used in the manuscript can predict reasonable value for interfacial shear stress between graphene layers. As noted in Fig. 2a in the manuscript, the value of λ is different from two layers to three layers of graphene membrane. How to understand this difference if assuming the shear between graphene layers is a constant?

Although the manuscript studied a detailed continuum mechanics problem, the implication to graphene/substrate adhesion towards graphene's engineering applications may be important. The manuscript can be further considered when the above question is addressed.

Reviewer #2:

Remarks to the Author:

The manuscript "Adhesion toughness of multilayer graphene membranes" by Wood, Harvey and Wang deals with a detailed model of circular blister test experiments under two different loading conditions with multilayer graphene membranes. Their model is based on partition of mixed-mode fractures and accounts for not only mode I and mode II fracture but also sliding between the graphene layers in multilayer graphene. They compared their theory with uniform pressure load blister test experiments done by Koenig et al.¹ They also obtained intrinsic mode I and mode II adhesion toughness values using Koenig et al.'s data and showed that the intrinsic adhesion toughness is independent of the number of layers, n even as the measured apparent adhesion toughness depended on n . Using these values, they estimated the apparent adhesion toughness in the case of point load blister test and compared it with Zong et al.'s measurements.² They get good agreement with the experiments in both cases.

Major Concerns

1. The paper focusses heavily on data from Koenig et al.'s work while ignoring other similar works in the literature. For example, the authors could have also looked at the follow-up work from the same group of researchers with the same or similar experimental setup.^{3,4} In these follow up papers, the adhesion toughness values were found to be lower than the one's in Koenig et al for both monolayer and multilayer graphene membranes. It would be really interesting to see what the model presented in this paper can tell about these other results.
2. In the supplementary text, the authors go from equation S1 to S20 by simply setting the width of the beam to a unit value. It is not quite clear how this can be done given that equation S1 is for a beam while equation S20 is supposed to be for a circular plate. It would be helpful if the authors can elaborate on this.
3. The author's need to elaborate on sliding of layers in multilayer graphene. When they say sliding, it is not quite clear if it is restricted to the delaminated region or extends to adhered region too. Also, can they comment for each case how the analysis changes, if it does?

Minor Concerns

1. Please check the units in the axis labels for δ and R_B in Fig. 2.
2. The following are not quite concerns but might be regarded as suggestions on how the authors can extend the current analysis and would be helpful if they can briefly comment on the same in the paper.
 - a. It would be interesting to see if the model can be extended to include sliding between graphene and the substrate itself which has been seen in some experiments. For example, see Kitt et al.⁵
 - b. It can be shown that sliding of a membrane on the substrate leads to symmetry breaking i.e. wrinkling. Can the current analysis be somehow extended to incorporate such complex behavior?

References

1. Koenig, S. P., Boddeti, N. G., Dunn, M. L. & Bunch, J. S. Ultrastrong adhesion of graphene membranes. *Nat. Nanotechnol.* **6**, 543–6 (2011).
2. Zong, Z., Chen, C.-L., Dokmeci, M. R. & Wan, K. Direct measurement of graphene adhesion on silicon surface by intercalation of nanoparticles. *J. Appl. Phys.* **107**, 26104 (2010).
3. Boddeti, N. G. *et al.* Mechanics of Adhered, Pressurized Graphene Blisters. *J. Appl. Mech.* **80**, 40909 (2013).
4. Boddeti, N. G. *et al.* Graphene blisters with switchable shapes controlled by pressure and adhesion. *Nano Lett.* **13**, 6216–21 (2013).
5. Kitt, A. L. *et al.* How graphene slides: measurement and theory of strain-dependent frictional forces between graphene and SiO₂. *Nano Lett.* **13**, 2605–10 (2013).

Reviewer #3:

Remarks to the Author:

This manuscript proposes a new understanding on the adhesion energy measurements of graphene by others (including Koenig et al and Zong et al). Unlike previous works, the authors in this work propose that the effect of mode mixity for interfacial fracture (or delamination) in the blister tests is the reason for different values of adhesion energy (or “adhesion toughness” as the authors would prefer) obtained for monolayer and multilayer graphene membranes by Koenig et al. The proposed method to predict the adhesion toughness seems to work very well in comparison with a slightly different experiment by Zong et al. based on point loaded blisters. While the effect of mode mixity is in general well known for adhesion or interfacial fracture of thin films and membranes, which has also been noted recently for graphene (e.g., Z. Cao, et al., “Mixed-mode interactions between graphene and substrates by blister tests”, *Journal of Applied Mechanics* 82, 081008, 2015, and “Mixed-mode traction-separation relations between graphene and copper by blister tests”, *Int. J. Solids and Structures* 84, 147-159, 2016), the present work is the first to interpret the data by Koenig et al in such a way that the difference between monolayer and multilayer graphene membranes results from the effect of mode mixity. As such, the authors have obtained two intrinsic values of adhesion toughness for mode I (opening) and mode II (shearing) respectively. This is quite remarkable because the mode II adhesion toughness of graphene has been extremely difficult to measure or calculate. On the other hand, the obtained adhesion toughness for mode I appears to be consistent with other experiments and theoretical predictions (e.g., W. Gao, et al., “Interfacial adhesion between graphene and silicon dioxide by density functional theory with van der Waals corrections”, *J. Phys. D: Appl. Phys.* 47, 255301, 2014).

Overall this reviewer favors publication of this work, but not without questions or comments for the authors to clarify. First of all, the authors should comment on a closely related work by Boddeti et al (“Mechanics of adhered, pressurized graphene blisters”, *J. Appl. Mech.* 80, 040909, 2013), in which the same experimental method as Koenig et al was used but the results are quite different. Can it be explained by the effect of mode mixity? If not, what else may be at play? The authors may also refer to a recent review paper for some discussions on this matter: D. Akinwande, et al., “A Review on Mechanics and Mechanical Properties of 2D Materials - Graphene and Beyond”, *Extreme Mechanics Letters* 13, 42-72 (2017).

The authors define the fracture mode mixity as G_I/G_{II} , which differs from the common definition used in fracture mechanics literature. More commonly, we would define the mode mixity as G_{II}/G_I , so that the mode I has the lowest mode mixity and typically the lowest toughness as well. It appears unnecessary to make a difference here.

There seems to a typo in the reference number for He et al in Page 2. Should be 5 instead of 4, according to the reference list at the end. However, Zong et al., listed as reference no. 4, is not mentioned until much later in the text. Perhaps the reference list should be re-ordered.

In Page 3, the authors claim that “fracture mode mixity and the sliding effect are not considered anywhere in the current mechanical models”. However, as noted above, the effect has been considered by Cao et al (although for different experiments) and discussed in the review paper by Akinwande et al.

Equation (1) looks similar to that of Hencky's solution as used by Koenig et al and that of a simpler solution by Yue et al (Ref. 8), but the dependence on Poisson's ratio in Eq. (2) is different. Please briefly comment on the reason for this difference.

A major question is concerning the partition of the energy release rate into mode I and mode II as given in Eqs. 7 and 8, with more details in the SI. The method is based on the authors' previous works (Refs. 14-18). First of all, this method has not been well accepted by others in the field, and there appears to be serious dispute over its validity. Second, with all information provided by the authors (including SI), it is very difficult to judge the soundness of this partition for anyone who is

not specialized in this particular problem (including this reviewer). One would have to go to the authors' previous papers to dig out the essence behind the equations. Even so (as this reviewer did), it is not an easy task. To the best of this reviewer's effort, this partition seems questionable if not flawed. To convince the readers, the authors would have to deliver the message more effectively. For example, how do they define G_I and G_{II} (before they reached Eqs. S1 and S2)? It is well known that the stress field for an interfacial crack in general cannot be separated into mode I and mode II (unless a cohesive zone model is used).

Another question is concerning the parameter λ in Eq. 7, which is said to represent the effect of sliding. It is unclear how this effect is taken into account. Eq. (S39) defines the parameter following some lengthy mathematics (which is very difficult to follow through), but it does not help understanding how the effect of sliding is treated. Is there any relative sliding displacement between the layers (ahead of the crack tip)? Is there a critical shear stress for sliding to occur?

Moreover, a correction factor $S(n)$ is introduced in Eq. 10 and given in Eq. 13. This appears to be completely ad hoc, in order to match the experimental data. This again casts some doubt on the partition method.

Is Eq. 22 true for all cases? It seems to depend on the parameter λ , which in turn depends on the number of layers.

Response to reviewers:

Reviewer 1

1.1. The authors need to correlate the physical mechanism of sliding to the parameter lambda in the manuscript, which was introduced as a dimensionless parameter indicating the level of sliding.

- **In the main article on p. 5:** The λ parameter in equations (7) and (9) represents the interlayer shearing and sliding effect at the blister crack tip, which is given as

$$\lambda = \bar{\lambda}S(n) \quad (10)$$

- **In the main article on p. 6:** In the case of monolayer graphene films, the shear force in equation (5) makes no contribution to the ERR in the membrane limit because there is no interlayer shearing and sliding. In the case of multilayer graphene films, interlayer shearing and sliding occurs near the blister crack tip, caused by the transition from membrane stretching to combined bending, stretching and through-thickness shearing. Consequently, interlayer shearing and sliding activates the shear force in equation (5). Its action is introduced through the λ parameter in conjunction with the interlayer shearing and sliding factor $S(n)$, which is assumed to take the following form:

$$S(n) = 1 - e^{1-n} \quad (13)$$

A more thorough and detailed explanation for the origin of λ is given in the Supplementary Information.

- **In the supplementary information on pp. 9–11:** The origin of the parameter $\bar{\lambda}$ is obvious, but the origin of the factor $S(n)$ needs to be explained. Monolayer films are considered first. First, some pertinent observations: As mentioned earlier, Jensen's^{24, 25} total ERR results for monolayer membranes agree very well with Wang & Tong's²⁹ values from finite element simulations. Jensen, however, only included the contributions from the bending moment M_B and the axial force N_B , as given by equations (S43) and (S44) respectively. This indicates that the through-thickness shear force P_B , given by equation (S45), does not contribute to ERR for monolayer membranes. Furthermore, in other work by the authors¹⁷,

two scenarios are considered using the methodology developed in the present work: (1) linear bending of monolayer films at small deflection, including the through-thickness force P_B , and (2) membrane stretching of monolayer films at large deflection without including the through-thickness force P_B . The analytical predictions for the adhesion toughness between photoresist films and copper substrates are in excellent agreement with the experimental results²⁶. The thicknesses of the photoresist films are 10 μm (for membrane stretching at large deflection), and 31 μm and 60 μm (both for linear bending at small deflection). This indicates that the through-thickness shear force has no effect on ERR for membrane stretching of monolayer films at large deflection, while it does have effect on ERR for linear bending of monolayer films at small deflection. Now, the explanation for these observations is given: In the case of linear bending at small deflection, through-thickness shear strain is produced by the through-thickness shear force. They together result in through-thickness shear strain energy and contribute to the ERR at crack tip. In the case of membrane stretching at large deflection, there is no through-thickness shear force in the membrane blister resulting in no through-thickness strain. Although transition from membrane stretching to combined bending, stretching and through-thickness shearing occurs near the crack tip, the through-thickness shear strain energy at crack tip is still negligible. Therefore, the through-thickness shear force has no effect on ERR in the membrane limit for monolayer films.

Multilayer graphene membranes¹⁸ are considered next. As before, for linear bending at small deflection, through-thickness shear force exists and produces through-thickness strain, resulting in extra ERR. In the membrane limit, it is expected that multilayer graphene films in the membrane region of a blister behave as a single layer because there is only membrane stretching. The transition from membrane stretching to combined bending, stretching and through-thickness shearing occurs near the crack tip. If a multilayer graphene film still behaves as a single layer in the transition region, as is the case for monolayer graphene membranes, the through-thickness shearing strain energy near the crack tip is still negligible, resulting in no ERR contribution; however, the transition can cause interlayer shearing and sliding in the transition region, at the crack tip in particular, the through-thickness shearing strain energy near the crack tip is no longer negligible, which does result in an ERR contribution. In fact, Koenig et al.'s¹⁸ observations indeed demonstrate that for a typical interlayer shearing and sliding effect, the adhesion toughness has a large decrease between monolayer and two-layer graphene films, but then remains fairly constant afterwards.

The present work takes the interlayer shearing and sliding near crack tip into consideration by introducing the interlayer shearing and sliding factor $S(n)$. The arguments for the proposed expression for $S(n)$ in equation (S51) are as follows: (1) Obviously, no interlayer shearing and sliding can exist in monolayer graphene membranes, so $S(1)=0$. (2) The factor $S(n)$ must account for the fact mentioned above that adhesion toughness has a large decrease between monolayer and two-layer graphene films and remains fairly constant afterwards¹⁸. (3) From the view point of continuum mechanics, the converged value of $S(n)$ for multilayer graphene films is assumed to make a complete transition from membrane stretching to combined bending, stretching and shearing at the crack tip, so $S(\infty)=1$. The validity of $S(n)$ in equation (S51) is tested by experimental results in the main article.

The values of the λ parameter, based on Koenig et al.s¹⁸ measurements, are recorded in Tables S1–S15. There is a large decrease from monolayer to two-layer graphene films and then only a small decrease from two-layer to three-layer graphene films. For the three-, four- and five-layer graphene films, the values of the λ parameter are very close to each other. This shows the typical interlayer shearing and sliding behaviour.

1.2. The authors need to confirm the values of lambda used in the manuscript can predict reasonable value for interfacial shear stress between graphene layers.

- This comment is related to Comment 1.1 as two perspectives on the same issue: Comment 1.1 relates λ and the interlayer shearing and sliding while this comment relates to λ and the interlayer shear stress. Therefore, please **refer to our response to Comment 1.1**.
- **In the supplementary information on p. 11:** In addition to above explanation of $S(n)$, further interpretation of its mechanical meaning may be useful. The average through-thickness shear stress at crack tip is $\tau_s = P_B/(nt)$ and the effective average through-thickness shear strain at crack tip due to interlayer shearing and sliding is assumed to be γ_s , which can be estimated from the equation below.

$$c_I \left(\frac{P_B S(n)}{\beta_{3-2D}} \right)^2 = \frac{1}{2} P_B \gamma_s \quad (\text{S52})$$

The left-hand side is the ERR from equation (S26) with through-thickness shear force acting alone with the introduction of $S(n)$. The right-hand side is the effective through-thickness shear strain energy in a volume of dimensions $1 \times 1 \times nt$. The effective through-thickness shear strain γ_s is then obtained as

$$\gamma_s = 2c_l \left(\frac{S(n)}{\beta_{3-2D}} \right)^2 P_B \quad (\text{S53})$$

and the effective through-thickness shear modulus G_s is then estimated as

$$G_s = \frac{P_B}{nt\gamma_s} \quad (\text{S54})$$

Substituting equations (S30), (S42) and (S53) into equation (S54) gives

$$G_s = \frac{0.1355E}{(1-\nu^2)S^2(n)} \quad (\text{S55})$$

The value 0.1355 is $\kappa(\gamma)$ in equation (S23) when $\gamma \rightarrow \infty$. It is seen that the mechanical meaning of $S(n)$ is to introduce an effective through-thickness shear modulus G_s to account for the interlayer shearing and sliding. Note that G_s is just an effective value instead of the actual material property. This is similar to case of classical plate theory in which the effective through-thickness shear modulus is infinitely large while the actual material property is finite.

- As noted in the bullet point above, the λ parameter represents the general averaged effect of interlayer shearing and sliding, but the interlayer shear stress is a local quantity, and in particular, the shear stress varies from interlayer to interlayer. Consequently, the present approach is unable to calculate the shear stress.
- **In the supplementary information on p. 11:** Finally, the critical interlayer shear stress for sliding is beyond the scope of the present work; however, the present methodology can be used to determine the mode I and II toughness between graphene layers using the blister test. The mode II toughness is considered to be the sliding toughness.

1.3. As noted in Fig. 2a in the manuscript, the value of lambda is different from two layers to three layers of graphene membrane. How to understand this difference if assuming the shear between graphene layers is a constant?

The changing value of λ is understood through the physical mechanism of shearing and sliding λ (which is the subject of Comment 1.1., above), which is has now been more thoroughly explained in the **supplementary information on pp. 9–11**. See the **response given to Comment 1.1., above**.

- **In the main article on p. 1 (Abstract):** Here we show that the presence of interlayer shearing and sliding near the blister crack tip, caused by the transition from membrane stretching to combined bending, stretching and through-thickness shearing, decreases the fracture mode mixity G_{II}/G_I , leading to a decrease in adhesion toughness.
- **In the main article on p. 2 (Introduction):** The present work shows that adhesion toughness is mode mixity-dependent, and that interlayer shearing and sliding near the blister crack tip, caused by the transition from membrane stretching to combined bending, stretching and through thickness shearing, decreases the mode mixity G_{II}/G_I , consequently reducing the adhesion toughness G_c .
- **In the main article on p. 15 (Conclusions):** The transition from membrane stretching to combined bending, stretching and through-thickness shearing near the blister crack tip causes interlayer shearing and sliding in multilayer graphene films.

Reviewer 2

2.1. The paper focusses heavily on data from Koenig et al.'s work while ignoring other similar works in the literature. For example, the authors could have also looked at the follow-up work from the same group of researchers with the same or similar experimental setup [3,4]. In these follow up papers, the adhesion toughness values were found to be lower than the one's in Koenig et al for both monolayer and multilayer graphene membranes. It would be really interesting to see what the model presented in this paper can tell about these other results.

- **In the main article on pp. 14–15:** A few other points are noteworthy. In recent work^{21,22} (following ref 1), Boddeti et al. reported further studies on the adhesion toughness between

monolayer graphene and silicon oxide substrates. The adhesion toughness was found to be $G_c = 0.24 \text{ J/m}^2$ which is significantly smaller than $G_c = 0.45 \text{ J/m}^2$, reported by Koenig et al.¹. Boddeti et al.²¹ suggest that the difference arises from the differences in interface properties such as roughness and chemical reactivity between the samples in ref 1 and the samples in refs 21,22. In line with this suggestion, the present work suggests that the reduction is caused by reduction of the mode I and mode II adhesion toughnesses at the interface, G_{Ic} and G_{IIc} , which can be estimated below. Taking the Young's modulus and Poisson's ratio still as $E = 1 \text{ TPa}$ and $\nu = 0.16$, equation (22) gives the mode II ERR components at failure as $G_{II} = 0.6986 \times 0.24 = 0.168 \text{ J/m}^2$. Then the mode I ERR component at failure is easily obtained as $G_I = 0.072 \text{ J/m}^2$. If the ratio between G_{Ic} and G_{IIc} is taken to be the same as that in ref 1, that is, $G_{IIc}/G_{Ic} = 2.896$, then G_{Ic} and G_{IIc} are then calculated to be $G_{Ic} = 0.130 \text{ J/m}^2$ and $G_{IIc} = 0.377 \text{ J/m}^2$. Clearly they are significantly smaller than $G_{Ic} = 0.230 \text{ J/m}^2$ and $G_{IIc} = 0.666 \text{ J/m}^2$ for the samples in ref 1. More information on adhesion toughness of graphenes can be found in the latest review paper²³.

21. Boddeti, N. G. et al. Mechanics of adhered, pressurized graphene blisters. *J. Appl. Mech.* **80**, 040909 (2013).
22. Boddeti, N. G. et al. Graphene blisters with switchable shapes controlled by pressure and adhesion. *Nano Lett.* **13**, 6216–6221 (2013).
23. Akinwande, D. et al. A review on mechanics and mechanical properties of 2D materials—Graphene and beyond. *Extreme Mech. Lett.* **13**, 42–77 (2017).

2.2. In the supplementary text, the authors go from equation S1 to S20 by simply setting the width of the beam to a unit value. It is not quite clear how this can be done given that equation S1 is for a beam while equation S20 is supposed to be for a circular plate. It would be helpful if the authors can elaborate on this.

- **In the supplementary information on p. 8:** Note that a circular blister is now specifically studied; therefore $b = 1$ as the forces in equations (S43)–(S45) are per unit width.
- **In the supplementary information on p. 1:** One example of 1D interface fracture is the circular blister fracture of thin films, which consists of only the mode I and II fracture modes. 1D interface fracture can be readily represented by a double cantilever beam (DCB) of unit width^{1,2}, as shown in Fig. S1. Its loading conditions consist of tip bending moments per unit

width, M_1 and M_2 , tip axial forces per unit width, N_1 and N_2 , and tip through-thickness shear forces per unit width, P_1 and P_2 .

- **In the supplementary information on p. 2:** [...]in which F_{op} and d_{op} represent the crack tip opening force per unit width and opening displacement respectively; F_{sh} and d_{sh} represent the crack tip interface shearing force per unit width and displacement respectively; and Δ represents the crack extension length.
- **In the supplementary information on pp. 1–7:** Equations (S26)–(S27) (formerly equations (S20) and (S21)) are given a much fuller explanation and derivation. The numbers of additions here is large, so they are not reproduced here.

2.3. The author’s need to elaborate on sliding of layers in multilayer graphene. When they say sliding, it is not quite clear if it is restricted to the delaminated region or extends to adhered region too. Also, can they comment for each case how the analysis changes, if it does?

- **Throughout the main article and the supplementary information,** the sliding is now referred to as “interlayer shearing and sliding near the blister crack tip”.
- **In the supplementary information on p. 11:** Note that the interface between graphene films and their substrates is assumed to be a rigid interface^{1–5,7–12}, that is, it is assumed that no relative shearing and sliding displacement occurs before separation. This is consistent with Koenig et al.’s¹⁸ work. The present methodology could, however, be extended to consider the shearing and sliding analytically by combining it with the authors’ mixed-mode partition theory for non-rigid interface fractures⁶. Some complex mechanical behaviour such as wrinkling³⁰ can be caused by this type sliding, which will be considered in future work.

2.4. Please check the units in the axis labels for δ and RB in Fig. 2.

The units of δ and R_B in Fig. 2e and f and Fig. 3 have been corrected to μm .

2.5. The following are not quite concerns but might be regarded as suggestions on how the authors can extend the current analysis and would be helpful if they can briefly comment on the same in the paper. a. It would be interesting to see if

the model can be extended to include sliding between graphene and the substrate itself which has been seen in some experiments. For example, see Kitt et al. [5] b. It can be shown that sliding of a membrane on the substrate leads to symmetry breaking i.e. wrinkling. Can the current analysis be somehow extended to incorporate such complex behavior?

- **For 2.5.a.: See response to comment 2.3. above.**
 - **For 2.5.b.: In the supplementary information on p. 11:** The present methodology could, however, be extended to consider the shearing and sliding analytically by combining it with the authors' mixed-mode partition theory for non-rigid interface fractures⁶. Some complex mechanical behaviour such as wrinkling³⁰ can be caused by this type sliding, which will be considered in future work.
30. Kitt, A. L. et al. How graphene slides: Measurement and theory of strain-dependent frictional forces between graphene and SiO₂. *Nano Lett.* **13**, 2605–2610 (2013).

Reviewer 3

*3.1. The authors should comment on a closely related work by Boddeti et al (“Mechanics of adhered, pressurized graphene blisters”, *J. Appl. Mech.* 80, 040909, 2013), in which the same experimental method as Koenig et al was used but the results are quite different. Can it be explained by the effect of mode mixity? If not, what else may be at play? The authors may also refer to a recent review paper for some discussions on this matter: D. Akinwande, et al., “A Review on Mechanics and Mechanical Properties of 2D Materials - Graphene and Beyond”, *Extreme Mechanics Letters* 13, 42-72 (2017).*

- **In the main article on pp. 14–15:** A few other points are noteworthy. In recent work^{21,22} (following ref 1), Boddeti et al. reported further studies on the adhesion toughness between monolayer graphene and silicon oxide substrates. The adhesion toughness was found to be $G_c = 0.24 \text{ J/m}^2$ which is significantly smaller than $G_c = 0.45 \text{ J/m}^2$, reported by Koenig et al.¹. Boddeti et al.²¹ suggest that the difference arises from the differences in interface properties such as roughness and chemical reactivity between the samples in ref 1 and the samples in refs 21,22. In line with this suggestion, the present work suggests that the reduction is caused by reduction of the mode I and mode II adhesion toughnesses at the

interface, G_{Ic} and G_{IIc} , which can be estimated below. Taking the Young's modulus and Poisson's ratio still as $E = 1 \text{ TPa}$ and $\nu = 0.16$, equation (22) gives the mode II ERR components at failure as $G_{II} = 0.6986 \times 0.24 = 0.168 \text{ J/m}^2$. Then the mode I ERR component at failure is easily obtained as $G_I = 0.072 \text{ J/m}^2$. If the ratio between G_{Ic} and G_{IIc} is taken to be the same as that in ref 1, that is, $G_{IIc}/G_{Ic} = 2.896$, then G_{Ic} and G_{IIc} are then calculated to be $G_{Ic} = 0.130 \text{ J/m}^2$ and $G_{IIc} = 0.377 \text{ J/m}^2$. Clearly they are significantly smaller than $G_{Ic} = 0.230 \text{ J/m}^2$ and $G_{IIc} = 0.666 \text{ J/m}^2$ for the samples in ref 1. More information on adhesion toughness of graphenes can be found in the latest review paper²³. The methodology developed above is also applied in the authors' recent work²⁴ to determine the mode I and mode II adhesion toughness of thin films by using blister tests. The analytical predictions agree very well with the experimental results reported by Cao et al.¹¹.

11. Cao, Z., Tao, L., Akinwande, D., Huang, R., & Liechti, K. M. Mixed-mode traction-separation relations between graphene and copper by blister tests. *Int. J. Solids. Struct.* **84**, 147–159 (2016).
21. Boddeti, N. G. et al. Mechanics of adhered, pressurized graphene blisters. *J. Appl. Mech.* **80**, 040909 (2013).
22. Boddeti, N. G. et al. Graphene blisters with switchable shapes controlled by pressure and adhesion. *Nano Lett.* **13**, 6216–6221 (2013).
23. Akinwande, D. et al. A review on mechanics and mechanical properties of 2D materials—Graphene and beyond. *Extreme Mech. Lett.* **13**, 42–77 (2017).
24. Harvey, C. M., Wang, S., Yuan, B., Thomson, R. C. & Critchlow, G. Determination of mode I and II adhesion toughness of thin films by circular blister tests. In preparation (2017).

3.2. The authors define the fracture mode mixity as G_{II}/G_I , which differs from the common definition used in fracture mechanics literature. More commonly, we would define the mode mixity as G_{II}/G_I , so that the mode I has the lowest mode mixity and typically the lowest toughness as well. It appears unnecessary to make a difference here.

The mode mixity has been redefined as $\rho = G_{II}/G_I$ throughout the manuscript and supporting information, and in the affected figures and tables.

3.3. There seems to a typo in the reference number for He et al in Page 2. Should be 5 instead of 4, according to the reference list at the end. However, Zong et al., listed as reference no. 4, is not mentioned until much later in the text. Perhaps the reference list should be re-ordered.

The reviewer is correct: References 4 and 5 were listed in the reverse order. This has been fixed now.

3.4. In Page 3, the authors claim that “fracture mode mixity and the sliding effect are not considered anywhere in the current mechanical models”. However, as noted above, the effect has been considered by Cao et al (although for different experiments) and discussed in the review paper by Akinwande et al.

- **In the main article on p. 2 (Introduction):** Note that the fracture mode mixity and the interlayer shear and sliding effect are not considered anywhere in the current analytical mechanical models^{1–10} and we argue that this has caused confusion when calculating adhesion toughness. Cao et al.^{11,12} did, however, recently report studies on adhesion toughness between photoresist films and copper substrates using blister tests and the finite element method. Two types of film are considered: One is pure photoresist film and the other is combined photoresist film and a monolayer graphene. Mode-mixity is considered by using cohesive zone modelling.

11. Cao, Z., Tao, L., Akinwande, D., Huang, R., & Liechti, K. M. Mixed-mode traction-separation relations between graphene and copper by blister tests. *Int. J. Solids. Struct.* **84**, 147–159 (2016).
12. Cao, Z., Tao, L., Akinwande, D., Huang, R. & Liechti, K. M. Mixed-mode interactions between graphene and substrates by blister tests. *J. Appl. Mech.* **82**, 081008 (2015).

3.5. Equation (1) looks similar to that of Hencky’s solution as used by Koenig et al and that of a simpler solution by Yue et al (Ref. 8), but the dependence on Poisson’s ratio in Eq. (2) is different. Please briefly comment on the reason for this difference.

- **In the supplementary information on p. 8:** The centre deflection δ , and the Poisson’s ratio ν -dependent parameters, $f(\nu)$ and $\varphi(\nu)$, are given in equations (1), (2) and (6) in the main article.

Note that various expressions for $f(v)$ and $\phi(v)$ are reported in literature²⁷ due to different approximations being used in their derivations. Jensen's^{24,25} total ERR results, however, are very close to Hencky's²⁸, as shown in the main article. Furthermore, Jensen's^{24,25} total ERR results for monolayer membranes agree very well with Wang & Tong's²⁹ values from finite element simulations. Therefore, based on these considerations, Jensen's^{24,25} expressions for $f(v)$ and $\phi(v)$, along with equations (S43)–(S45) are used in the present work.

3.6. A major question is concerning the partition of the energy release rate into mode I and mode II as given in Eqs. 7 and 8, with more details in the SI. The method is based on the authors' previous works (Refs. 14-18). First of all, this method has not been well accepted by others in the field, and there appears to be serious dispute over its validity. Second, with all information provided by the authors (including SI), it is very difficult to judge the soundness of this partition for anyone who is not specialized in this particular problem (including this reviewer). One would have to go to the authors' previous papers to dig out the essence behind the equations. Even so (as this reviewer did), it is not an easy task. To the best of this reviewer's effort, this partition seems questionable if not flawed. To convince the readers, the authors would have to deliver the message more effectively. For example, how do they define G_I and G_{II} (before they reached Eqs. S1 and S2)? It is well known that the stress field for an interfacial crack in general cannot be separated into mode I and mode II (unless a cohesive zone model is used).

- **In the supplementary information on p. 1:** During the last decade or so, the authors and their colleagues have developed an orthogonal pure mode partition methodology for partitioning mixed-mode 1D interface fractures in layered composite materials into their pure mode components. One example of 1D interface fracture is the circular blister fracture of thin films, which consists of only the mode I and II fracture modes. 1D interface fracture can be readily represented by a double cantilever beam (DCB) of unit width^{1,2}, as shown in Fig. S1. Its loading conditions consist of tip bending moments per unit width, M_1 and M_2 , tip axial forces per unit width, N_1 and N_2 , and tip through-thickness shear forces per unit width, P_1

and P_2 . Figure S1b shows the internal loads at the crack tip and the sign convention of the interface normal stress σ_n and shear stress τ_s . Extensive analytical and numerical studies have been carried out to prove the validity of the methodology³⁻¹² and various independent experimental tests results have been used to assess the methodology¹²⁻¹⁷. It is found to be sound and the development is clear and thorough. A detailed explanation of the methodology³⁻¹²—even just the aspects that are closely-related to the present work—is not possible here. Therefore, in order to focus on the present work, only the most essential parts are given for physical understanding in what follows.

- **In the supplementary information on pp. 1–3:** The former equations (S1) and (S2) have become equations (S7) and (S8), and the most essential parts of the explanation are given.

Based on virtual crack closure technique and linear elastic fracture mechanics, the mode I and mode II energy release rates (ERRs) can be written as

$$G_I = \lim_{\Delta \rightarrow 0} \frac{F_{op} d_{op}}{2\Delta} \quad (\text{S1})$$

$$G_{II} = \lim_{\Delta \rightarrow 0} \frac{d_{sh} F_{sh}}{2\Delta} \quad (\text{S2})$$

in which F_{op} and d_{op} represent the crack tip opening force per unit width and opening displacement respectively; F_{sh} and d_{sh} represent the crack tip interface shearing force per unit width and displacement respectively; and Δ represents the crack extension length. Equations (S1) and (S2) can be written in the following forms⁷⁻¹⁰ based on 2D elasticity:

$$G_I = c_I \left(M_{1B} - \frac{M_{2B}}{\beta_{1-2D}} - \frac{N_{1B}}{\beta_{2-2D}} - \frac{N_{2B}}{\beta_{3-2D}} - \frac{P_{1B}}{\beta_{4-2D}} - \frac{P_{2B}}{\beta_{5-2D}} \right) \times \left(M_{1B} - \frac{M_{2B}}{\beta'_{1-2D}} - \frac{N_{1B}}{\beta'_{2-2D}} - \frac{N_{2B}}{\beta'_{3-2D}} - \frac{P_{1B}}{\beta'_{4-2D}} - \frac{P_{2B}}{\beta'_{5-2D}} \right) \quad (\text{S3})$$

$$G_{II} = c_{II} \left(M_{1B} - \frac{M_{2B}}{\theta_{1-2D}} - \frac{N_{1B}}{\theta_{2-2D}} - \frac{N_{2B}}{\theta_{3-2D}} - \frac{P_{1B}}{\theta_{4-2D}} - \frac{P_{2B}}{\theta_{5-2D}} \right) \times \left(M_{1B} - \frac{M_{2B}}{\theta'_{1-2D}} - \frac{N_{1B}}{\theta'_{2-2D}} - \frac{N_{2B}}{\theta'_{3-2D}} - \frac{P_{1B}}{\theta'_{4-2D}} - \frac{P_{2B}}{\theta'_{5-2D}} \right) \quad (\text{S4})$$

By comparing equations (S3) and (S4) with equations (S1) and (S2), it is seen that the terms in the first and second brackets of equation (S3) correspond to F_{op} and d_{op} respectively, while the terms in the first and second brackets of equation (S4) correspond to d_{sh} and F_{sh} respectively. There are generally two sets of orthogonal pure mode I and II modes and they depend on the material properties, interface properties, fracture location, crack extension size, etc. The first set of 2D elasticity-based pure mode I and II modes is represented by θ_{i-2D} and β_{i-2D} respectively (with $i = 1,2,3,4,5$); and the second set by θ'_{i-2D} and β'_{i-2D} . The two sets are different from each other in the case of bi-material interfaces because the material mismatch causes a phase difference between variations of stress and displacement. In the case of homogeneous interfaces, the two sets of orthogonal pure modes coincide with each other and equations (S3) and (S4) become

$$G_I = c_I \left(M_{1B} - \frac{M_{2B}}{\beta_{1-2D}} - \frac{N_{1B}}{\beta_{2-2D}} - \frac{N_{2B}}{\beta_{3-2D}} - \frac{P_{1B}}{\beta_{4-2D}} - \frac{P_{2B}}{\beta_{5-2D}} \right)^2 \quad (S5)$$

$$G_{II} = c_{II} \left(M_{1B} - \frac{M_{2B}}{\theta_{1-2D}} - \frac{N_{1B}}{\theta_{2-2D}} - \frac{N_{2B}}{\theta_{3-2D}} - \frac{P_{1B}}{\theta_{4-2D}} - \frac{P_{2B}}{\theta_{5-2D}} \right)^2 \quad (S6)$$

Furthermore, in the case of isotropic materials, equations (S5) and (S6) reduce to

$$G_I = c_I \left(M_{1B} - \frac{M_{2B}}{\beta_{1-2D}} - \frac{N_{1Be}}{\beta_{2-2D}} - \frac{P_{1B}}{\beta_{3-2D}} - \frac{P_{2B}}{\beta_{4-2D}} \right)^2 \quad (S7)$$

$$G_{II} = c_{II} \left(M_{1B} - \frac{M_{2B}}{\theta_{1-2D}} - \frac{N_{1Be}}{\theta_{2-2D}} - \frac{P_{1B}}{\theta_{3-2D}} - \frac{P_{2B}}{\theta_{4-2D}} \right)^2 \quad (S8)$$

where $N_{1Be} = N_{1B} - N_{2B}/\gamma$. The pure modes, θ_{i-2D} and β_{i-2D} (with $i = 1,2,3,4$), in equations (S7) and (S8) were derived by the authors^{7,10} and have been thoroughly verified against numerical simulations.

- **In the supplementary information on pp. 4–5:** In the following, equations (S7) and (S8) are reduced to the case of thin films in the blister test to determine the adhesion toughness, for example, the adhesion toughness of multilayer graphene films¹⁸. The substrate is treated as infinitely thick and the films as very thin, as shown in Fig. S2a; therefore, the thickness ratio tends to infinity $\gamma \rightarrow \infty$. The authors' latest work on the mechanical

behaviour of thin film spallation^{14–17} shows that excellent agreement is achieved with experimental results^{19–23} when the material mismatch between a film and its substrate is neglected. Furthermore, in these studies^{14,15} slightly worse agreement was found with experimental results^{19,20} when the mismatch^{8,9} was taken into account. Therefore, the present work also neglects the material mismatch, and equations (S7) and (S8) become

$$G_I = c_I \left(M_B - \frac{N_B}{\beta_{2-2D}} - \frac{P_B}{\beta_{3-2D}} \right)^2 \quad (\text{S26})$$

$$G_{II} = c_{II} \left(M_B - \frac{N_B}{\theta_{2-2D}} - \frac{P_B}{\theta_{3-2D}} \right)^2 \quad (\text{S27})$$

where M_B , N_B and P_B , which are shown in Fig. S2b, are the effective crack tip bending moment, axial force and shear force respectively. The pure modes θ_{2-2D} and β_{2-2D} when $\gamma \rightarrow \infty$, based on Suo & Hutchinson^{1,2}, are

$$(\theta_{2-2D}, \beta_{2-2D}) = \left(-\frac{2.697}{h_1}, \frac{4.450}{h_1} \right) \quad (\text{S28})$$

where h_1 is the film thickness. Note that the authors' pure modes θ_{2-2D} and β_{2-2D} in equations (S11) and (S12) also give very close values to equation (S28); however, in the present work, equation (S28) is used.

- **In the supplementary information on p. 4:** It is worth noting that in the absence of through-thickness shear forces, P_{1B} and P_{2B} , equations (S7) and (S8) are extremely close to Suo & Hutchinson's 2D partitions^{1,2}; that is, the pure modes in equations (S9)–(S12) are nearly identical to Suo & Hutchinson's pure modes^{1,2} and are just presented in different forms.
- **In the supplementary information on p. 6:** Note that the pure modes in equation (S28) (refs. 1,2) are used in deriving equation (S37). When pure modes in equations (S11) and (S12), derived by the authors⁷, are used, the ratio becomes $\theta_{1-2D}/\beta_{1-2D} = -75/121 \approx -0.6198$ which is very close to -0.6059 in equation (S37). In the present work, equation (S37) is used as it is believed to be more accurate.

3.7. Another question is concerning the parameter λ in Eq. 7, which is said to represent the effect of sliding. It is unclear how this effect is taken into account. Eq. (S39) defines the parameter following some lengthy mathematics (which is very difficult to follow through), but it does not help understanding how the effect of sliding is treated. Is there any relative sliding displacement between the layers (ahead of the crack tip)? Is there a critical shear stress for sliding to occur?

- Reviewer 1 asks similar questions regarding λ . A **detailed response is given in our responses to comments 1.1, 1.2 and 1.3 (above)**.
- **In the supplementary information on p. 11:** Note that the interface between graphene films and their substrates is assumed to be a rigid interface^{1-5,7-12}, that is, it is assumed that no relative shearing and sliding displacement occurs before separation. This is consistent with Koenig et al.'s¹⁸ work. The present methodology could, however, be extended to consider the shearing and sliding analytically by combining it with the authors' mixed-mode partition theory for non-rigid interface fractures⁶. Some complex mechanical behaviour such as wrinkling³⁰ can be caused by this type sliding, which will be considered in future work.
- **In the supplementary information on p. 11:** Finally, the critical interlayer shear stress for sliding is beyond the scope of the present work; however, the present methodology can be used to determine the mode I and II toughness between graphene layers using the blister test. The mode II toughness is considered to be the sliding toughness.

3.8. Moreover, a correction factor $S(n)$ is introduced in Eq. 10 and given in Eq. 13. This appears to be completely ad hoc, in order to match the experimental data. This again casts some doubt on the partition method.

Reviewer 1 asks similar questions regarding λ (and consequently $S(n)$). A **detailed response is given in our response to comments 1.1, 1.2 and 1.3 (above)**.

3.9. Is Eq. 22 true for all cases? It seems to depend on the parameter λ , which in turn depends on the number of layers.

- **In the supplementary information on p. 7:** Furthermore, by combining either equations (8) and (15), or equations (19), (21) and $G_{II}(1 + 1/\rho) = G_J(1 + \eta)$, then

$$G_{II} = 0.6986G_J \quad (22)$$

- **Equation (22) is indeed true for all cases with Poisson's ratio $\nu = 0.16$ and does not depend on λ .** To see this, the third term in equation (15) gives $p\delta = G_J f(\nu)/\zeta(\nu)$, which when substituted into equation (8) gives

$$\frac{G_{II}}{G_J} = \frac{0.3373 f(\nu) (1.400 + 0.2358\nu)^2}{8\zeta(\nu) \varphi(\nu) f(\nu)}$$

which is only a function of ν . Substituting Poisson's ratio $\nu = 0.16$ gives equation (22).

- Alternatively consider equations (19), (21) and $G_{II}(1 + 1/\rho) = G_J(1 + \eta)$. This produces

$$\frac{G_{II}}{G_J} = \frac{1 + \eta}{1 + 1/\rho} = \frac{1 + 0.5580\lambda(1.470 + \lambda)}{1 + (0.7349 + \lambda)^2/1.252} = \frac{[1 + 0.5580\lambda(1.470 + \lambda)]}{1.4314[1 + 0.5580\lambda(1.470 + \lambda)]} = 0.6986$$

after simplification as the square brackets containing λ cancel. Note that equations (19) and (21) have already taken Poisson's ratio $\nu = 0.16$.

Reviewers' Comments:

Reviewer #1:

Remarks to the Author:

The current reviewer noticed that Reviewer #3 raised two important questions, and would like to add more comments.

(1) Reviewer #3 questioned the validity on the partition of energy release rate into mode I and mode II.

-- This theory is based on the authors' previous works, which are not familiar to the majority of scholars in mechanics field (including the current viewer and reviewer #3). Without significant effects, it is hard to justify the theory itself.

(2) Reviewer #3 suggested to deliver partition theory more effectively so that the audiences can follow the paper without dig into the authors' previous publications.

-- The authors have not fully address this comment. The authors did not give sufficient background to the theory. Instead, they simply claimed the theory itself is well established and verified, which may not be true. More importantly, it should be noticed that the audiences will not learn a lot from this work, especially those equations in the SI, without the necessary background. One important feature of Nature Comm is to attract broader audiences, and thus to make significant impact in related fields. However, the current manuscript may not sever this purpose well enough. The manuscript may be more suitable for specialized Journals, where the main audiences are more interested in partition theory. Essentially, this work is an application of such a theory.

Reviewer #2:

Remarks to the Author:

All this reviewer's concerns have been adequately addressed.

Response to reviewers:

Reviewer 1

1.1. Reviewer #3 questioned the validity on the partition of energy release rate into mode I and mode II. -- This theory is based on the authors' previous works, which are not familiar to the majority of scholars in mechanics filed (including the current viewer and reviewer #3). Without significant effects, it is hard to justify the theory itself.

- We have tried to convincingly demonstrate the validity of the partition theory in three ways: **(1)** By showing that the partition theory does in fact reduce to the widely-accepted partition theory of Suo and Hutchinson^{1,2} when just bending moments and axial forces are present with no shear forces. Therefore the partition theory does not dispute what is already widely-accepted to be valid. Suo and Hutchinson's^{1,2} work, however, does not deal with shear forces, but consideration of shear forces is crucial for this work on multilayer graphene films. The work in this manuscript builds on the foundation of the authors' existing partition theory to consider thin films in the blister test, and it is well-established to provide the framework for this extension. **(2)** By providing further explanations of the mechanical underpinnings of the partition theory that are accessible to readers who are not familiar with solid mechanics. *The additions to the manuscript that address this point are detailed below in the response to Comment 1.2.* **(3)** By directing interested readers to more-in-depth peer-reviewed publications in which the validity is tested.
- Furthermore, regarding the validity of the partition theory, the theory has been presented in several high quality international conferences as invited plenary lectures. The latest plenary lecture was presented in June 2017 at the 14th International Conference on Fracture (ICF14) which is regarded as the top conference on fracture mechanics, held every 4 years, and most world-leading fracture mechanics experts attended. In addition, several technical sessions were invited to be organised to report the work, including in ICF14, and in August 2017 at the 21st International Conference on Composite Materials (ICCM21).
- Additions (highlighted in yellow) have been made to the Supplementary Information as follows:

- **In the Supplementary Information on p. 3:** The pure modes, θ_{i-2D} and β_{i-2D} (with $i = 1,2,3,4$), in equations (S7) and (S8) were derived by the authors^{7,10} using a powerful orthogonal pure mode methodology and have been thoroughly verified against numerical simulations (interested readers are advised to read refs. 7 and 10). They are recorded below.
- **In the Supplementary Information on p. 5:** It is worth noting that in the absence of through-thickness shear forces, P_{1B} and P_{2B} , equations (S7) and (S8) are extremely close to Suo & Hutchinson's 2D partitions^{1,2}; that is, the pure modes in equations (S9)–(S12) are nearly identical to Suo & Hutchinson's pure modes^{1,2} and are just presented in different forms.
- **Furthermore, in the Supplementary Information on p. 7:** Note that the pure modes in equation (S28) (refs. 1,2) are used in deriving equation (S37). When pure modes in equations (S11) and (S12), derived by the authors⁷, are used, the ratio becomes $\theta_{1-2D}/\beta_{1-2D} = -75/121 \approx -0.6198$ which is very close to -0.6059 in equation (S37). In the present work, equation (S37) is used as it is believed to be more accurate.

References

1. Suo, Z. & Hutchinson, J. W. Interface crack between two elastic layers. *Int. J. Fract.* **43**, 1–18 (1990).
2. Hutchinson, J. W. & Suo, Z. Mixed mode cracking in layered materials. *Adv. Appl. Mech.* **92**, 63–191 (1992).

1.2. Reviewer #3 suggested to deliver partition theory more effectively so that the audiences can follow the paper without dig into the authors' previous publications. -- The authors have not fully address this comment. The authors did not give sufficient background to the theory. Instead, they simply claimed the theory itself is well established and verified, which may not be true. More importantly, it should be noticed that the audiences will not learn a lot from this work, especially those equations in the SI, without the necessary background. One important feature of Nature Comm is to attract broader audiences, and thus to make significant impact in related fields. However, the current manuscript may not sever this purpose well enough. The manuscript may be more suitable for

specialized Journals, where the main audiences are more interested in partition theory.

- We have added substantial further explanations (highlighted in yellow) to the Supplementary Information. We believe that a reader who is not a specialist in solid mechanics or fracture mechanics can follow the explanations of the partition theory, and understand why it takes the form it does in equations (S3) and (S4). We give the explanations with reference to the well-known and easy-to-understand virtual crack closure technique as it provides a simple ‘access point’ to understanding the partition theory based on energy considerations without needing to consider the in-depth solid mechanics. This makes it more accessible to a broader audience.
- We do not, however, give a full derivation of each of the terms in the partition theory, and we don’t believe it is necessary in order to ‘access’ what is presented in the rest of the manuscript and Supplementary Information. The terms are instead simply presented in equations (S9) to (S25). The alternative would be to present full derivations in the manuscript and we don’t think this would be appropriate – this would be a separate much larger work. Instead we direct interested readers to refs. 7 and 10.
- Nevertheless, additions are made to the Supplementary Information as follows:
- **In the Supplementary Information on p. 1 to 3:** Based on the well-known virtual crack closure technique and linear elastic fracture mechanics, the mode I and mode II energy release rates (ERRs) can be written as

$$G_I = \lim_{\Delta \rightarrow 0} \frac{F_{op} d_{op}}{2\Delta} \quad (\text{S1})$$

$$G_{II} = \lim_{\Delta \rightarrow 0} \frac{d_{sh} F_{sh}}{2\Delta} \quad (\text{S2})$$

in which F_{op} and d_{op} represent the crack tip opening force per unit width and opening displacement respectively; F_{sh} and d_{sh} represent the crack tip interface shearing force per unit width and displacement respectively; and Δ represents the crack extension length. Equations (S1) and (S2) can be written in the following forms⁷⁻¹⁰ based on 2D elasticity:

$$G_I = c_I \left(M_{1B} - \frac{M_{2B}}{\beta_{1-2D}} - \frac{N_{1B}}{\beta_{2-2D}} - \frac{N_{2B}}{\beta_{3-2D}} - \frac{P_{1B}}{\beta_{4-2D}} - \frac{P_{2B}}{\beta_{5-2D}} \right) \quad (S3)$$

$$\times \left(M_{1B} - \frac{M_{2B}}{\beta'_{1-2D}} - \frac{N_{1B}}{\beta'_{2-2D}} - \frac{N_{2B}}{\beta'_{3-2D}} - \frac{P_{1B}}{\beta'_{4-2D}} - \frac{P_{2B}}{\beta'_{5-2D}} \right)$$

$$G_{II} = c_{II} \left(M_{1B} - \frac{M_{2B}}{\theta_{1-2D}} - \frac{N_{1B}}{\theta_{2-2D}} - \frac{N_{2B}}{\theta_{3-2D}} - \frac{P_{1B}}{\theta_{4-2D}} - \frac{P_{2B}}{\theta_{5-2D}} \right) \quad (S4)$$

$$\times \left(M_{1B} - \frac{M_{2B}}{\theta'_{1-2D}} - \frac{N_{1B}}{\theta'_{2-2D}} - \frac{N_{2B}}{\theta'_{3-2D}} - \frac{P_{1B}}{\theta'_{4-2D}} - \frac{P_{2B}}{\theta'_{5-2D}} \right)$$

By comparing equation (S3) with equation (S1), it is seen that the terms in the first and second brackets of equation (S3) correspond to F_{op} and d_{op} respectively, which are linearly proportional to the crack tip loads. This is required by linear elastic fracture mechanics. The crack tip loads consist of the bending moments per unit width, M_{1B} and M_{2B} , the axial forces per unit width, N_{1B} and N_{2B} , and the through-thickness shear forces per unit width, P_{1B} and P_{2B} . The coefficients c_I and c_{II} are constants. The parameters β_{i-2D} and β'_{i-2D} (with $i=1,2,3,4,5$) are independent of the crack tip loads and dependent on the DCB material properties, interface properties, fracture location, crack extension size, etc. They are called pure mode II modes for reasons best shown by example: When the crack tip loading conditions are $M_{1B} = 1$, $M_{2B} = \beta_{1-2D}$, $N_{1B} = 0$, $N_{2B} = 0$, $P_{1B} = 0$, $P_{2B} = 0$, or in a vector form $\{M_{1B} \ M_{2B} \ N_{1B} \ N_{2B} \ P_{1B} \ P_{2B}\}^T = \{1 \ \beta_{1-2D} \ 0 \ 0 \ 0 \ 0\}^T$ with the superscript T denoting transposition, the first bracket in equation (S3), corresponding to F_{op} , equals zero and therefore mode I ERR $G_I = 0$; hence, β_{1-2D} is called a pure mode II mode due to zero crack tip opening force. Similarly, β_{i-2D} (with $i=2,3,4,5$) are also called pure mode II modes due to zero crack tip opening force. Using the equivalent explanation, β'_{i-2D} (with $i=1,2,3,4,5$) are called pure mode II modes due to zero crack tip opening displacement. It is worth noting that β_{i-2D} and β'_{i-2D} (with $i=1,2,3,4,5$) are different from each other in the case of bi-material interfaces because the material mismatch causes a phase difference between the variations of stress and displacement^{8,9,18}, and they are also crack tip extension size-dependent^{8,9,18}.

Similarly, By comparing equation (S4) with equation (S2), it is seen that the terms in the first and second brackets of equation (S4) correspond to d_{sh} and F_{sh} respectively. By using

the same explanation as above, θ_{i-2D} (with $i=1,2,3,4,5$) are called pure mode I modes due to zero crack tip shearing displacement, and θ'_{i-2D} (with $i=1,2,3,4,5$) are called pure mode I modes due to zero crack tip shearing force. Again, θ_{i-2D} and θ'_{i-2D} (with $i=1,2,3,4,5$) are different from each other in the case of bi-material interfaces because the material mismatch causes a phase difference between variations of stress and displacement^{8,9,18} and they are also crack tip extension size dependent^{8,9,18}. In the case of homogeneous interfaces, β_{i-2D} and β'_{i-2D} (with $i=1,2,3,4,5$) are equal to each other, and θ_{i-2D} and θ'_{i-2D} (with $i=1,2,3,4,5$) are also equal to each other because then there is no phase difference between variations of stress and displacement^{8,9,18} and they are also independent of crack tip extension size^{8,9,18}. Then, equations (S3) and (S4) become

$$G_I = c_I \left(M_{1B} - \frac{M_{2B}}{\beta_{1-2D}} - \frac{N_{1B}}{\beta_{2-2D}} - \frac{N_{2B}}{\beta_{3-2D}} - \frac{P_{1B}}{\beta_{4-2D}} - \frac{P_{2B}}{\beta_{5-2D}} \right)^2 \quad (S5)$$

$$G_{II} = c_{II} \left(M_{1B} - \frac{M_{2B}}{\theta_{1-2D}} - \frac{N_{1B}}{\theta_{2-2D}} - \frac{N_{2B}}{\theta_{3-2D}} - \frac{P_{1B}}{\theta_{4-2D}} - \frac{P_{2B}}{\theta_{5-2D}} \right)^2 \quad (S6)$$

References

18. Sun, C. T. & Jih, C. J. On strain energy release rates for interfacial cracks in bi-material media. *Eng. Fract. Mech.* **28**, 13–20 (1987).

1.3. Essentially, this work is an application of such a theory.

- The reviewer says that the work is essentially just an application of the partition theory. Actually the manuscript presents much more than just an application of the partition theory. Section S2 in the Supplementary Information starts with equations (S7) and (S8), which represent the authors' partition theory. Before it can be applied to the case of thin films in the blister test, it needs to be *extended* to treat the limiting case of thin films. It is seen from Section S2 that this is not a straightforward reduction and requires careful consideration of energy release rate, pure modes, and the orthogonality relationship between pure modes. Actually, the Supplementary Information derives (for the first time) a generalisation of partition theory for thin layers, which is required before it can be applied to the blister test. To clearly show this essential development (for the first time), the section title is changed to be **S2. Development of mixed-mode partition theory for thin film blister test.**

- Furthermore, the aim of this work was to develop a mechanical model and methodology to correctly determine the adhesion toughness between multilayer graphene films and substrates. Even if the work was just an application of the partition theory (which it isn't), the aim is nevertheless achieved, and this even on its own, we believe, would make it a valuable contribution to the field. The thoroughly-derived analytical formulae and methodology can, for the first time, be readily used and accessed by broader audiences. We therefore believe it will make a significant impact in the field.
- **In the Supplementary Information on p. 5:** In the following, equations (S7) and (S8) are extended to the case of thin films in the blister test to determine the adhesion toughness, for example, the adhesion toughness of multilayer graphene films¹⁹.

Reviewer 2

2.1. All this reviewer's concerns have been adequately addressed.

Reviewer 3 (from the first round of review)

3.1. Overall this reviewer favors publication of this work...

- Although Reviewer 3 was not able to comment on our first revision, he/she was in favour of publication subject to our clarifications. Reviewer 3 actually made a very thorough first review and made many comments which we then made our best effort to answer. Furthermore, in response to Reviewer 1 in this second round of review, we have added yet further clarification on some of these points (see above).

Reviewers' Comments:

Reviewer #1:

Remarks to the Author:

All comments have been adequately addressed.